# Monitoring of Unhatched Eggs in Hermann’s Tortoise (*Testudo hermanni*) after Artificial Incubation and Possible Improvements in Hatching

**DOI:** 10.3390/ani11020478

**Published:** 2021-02-11

**Authors:** Alenka Dovč, Mateja Stvarnik, Renata Lindtner Knific, Gordana Gregurić Gračner, Igor Klobučar, Olga Zorman Rojs

**Affiliations:** 1Institute for Poultry, Birds, Small Animals and Reptiles, Veterinary Faculty, University of Ljubljana, Gerbičeva 60, 1000 Ljubljana, Slovenia; Renata.LindtnerKnific@vf.uni-lj.si (R.L.K.); olga.zormanrojs@vf.uni-lj.si (O.Z.R.); 2Clinic for Reproduction and Large Animals, Veterinary Faculty, Gerbičeva 60, 1000 Ljubljana, Slovenia; mateja.stvarnik@vf.uni-lj.si (M.S.); igor.klobucar@vf.uni-lj.si (I.K.); 3Department for Hygiene, Faculty of Veterinary Medicine, University of Zagreb, Behaviour and Animal Welfare, Heinzelova 55, 10000 Zagreb, Croatia; ggracner@vef.hr

**Keywords:** *Testudo hermanni*, eggs, artificial incubation, hatching artefacts

## Abstract

**Simple Summary:**

Hermann’s tortoises (*Testudo hermanni*) are common and popular pets. Their life expectancy is over 80 years. The major threats to these tortoises include loss of natural habitat, pollution, urbanization, tourism, road deaths, the pet trade, and possible transmission of various infectious diseases. Health problems can begin during artificial incubation and last a lifetime. For this reason, it is necessary to breed healthy hatchlings, and tortoise welfare will increase in parallel. Conditions for hatching and control of egg failure must be controlled. The presence of unfertilized and infected eggs in the lowest possible numbers is important for successful hatching. These parameters can be improved by monitoring breeding parents. Proper hatching will keep many tortoises alive and allow them to live long healthy lives and is essential for tortoise welfare allowing a more economical method of breeding. In this study, the causes of embryonic mortality in Hermann’s tortoises (*Testudo hermanni*) during artificial incubation on a large farm in Slovenia were determined. Possible improvements in Hermann’s tortoise hatching and methods to increase animal welfare were described.

**Abstract:**

The causes of embryonic mortality in Hermann’s tortoises (*Testudo hermanni*) during artificial incubation were determined. Total egg failure at the end of the hatching period was investigated. The hatching artefacts represented 19.2% (N = 3557) of all eggs (N = 18,520). The viability rate of incubated eggs was 80.8%. The eggs, i.e., embryos, were sorted according to the cause of unsuccessful hatching and subsequently analyzed. Some of the eggs were divided into two or more groups. Unfertilized eggs were confirmed in 61.0%, infected eggs in 52.5%, and eggs in various stages of desiccation in 19.1%. This group also included mummified embryos. *Pseudomonas aeruginosa*, *Bacillus* sp., *Purpureocillium lilacinum*, and *Escherichia coli* were frequently confirmed in infected eggs. Embryos were divided into three groups: embryos up to 1.0 cm—group 1 (2.2%), embryos from 1.0 cm to 1.5 cm—group 2 (5.4%) and embryos longer than 1.5 cm—group 3 (7.3%) of all unhatched eggs. Inability of embryos to peck the shell was found in 1.3%. These tortoises died shortly before hatching. Embryos still alive from the group 2 and group 3 were confirmed in 0.7% of cases. Dead and alive deformed embryos and twins were detected in the group 3 in 0.5% and 0.1% of cases, respectively. For successful artificial hatching, it is important to establish fumigation with disinfectants prior to incubation and elimination of eggs with different shapes, eggs with broken shells, and eggs weighted under 10 g. Eggs should be candled before and periodically during artificial incubation, and all unfertilized and dead embryos must be removed. Heartbeat monitor is recommended. Proper temperature and humidity, incubation of “clean” eggs on sterile substrate and control for the presence of mites is essential. Monitoring of the parent tortoises is also necessary.

## 1. Introduction

Hermann’s tortoise is a medium-sized terrestrial species that occurs naturally in the European Mediterranean. Two subspecies are currently distinguished: *Testudo hermanni hermanni* in Western Europe and *Testudo hermanni boettgeri* (*T. h. boettgeri*) in Eastern Europe. Eggs are white, hard-shelled, and almost elliptical in shape. *Testudo hermanni hermanni* lays a maximum of seven eggs per clutch, and *T. h. boettgeri* lays a maximum of nine eggs [1].

The yolk content ranges from 32% to 55% in turtles and from 72% to 99% in lizards and snakes. Reptile eggs do not contain chalaza that hold the yolk in a central position. The developing embryo floats on the surface of the yolk. In the case of incorrect movement or rotation of the egg, the embryonic vessels may rupture and consequently cause embryonic death [2]. Contrary to bird eggs, it is generally assumed that reptile eggs do not respond well if rolled or turned early on or after laying [3]. Stvarnik et al., 2017 [4] found the egg of *T. h. boettgeri* was composed of 12.5% shell, 46.9% albumen, and 40.6% yolk. The refractive index was 1.3341 in the albumen and 1.5120 in the yolk. The albumen contained 98.2% water, 0.9% protein, 0.7% ash, and traces of fat. The yolk contained 60.6% water, 21.0% protein, 14.2% fat, and 4.0% ash.

Hermann’s tortoises are active from March until the end of October, and their nesting activities peak during March–April and August–September [5]. The minimum size of the plastron is approximately 120 mm for sexually active males and 130 mm for females. Tortoises are promiscuous in their mating habits, and there is no evidence of mate selection in relation to size. In the nature, nesting site selection correlates with ground temperature, and nesting itself occurs mainly in the early evening before nightfall when ground temperature characteristics are most indicative of the relative temperature at a given site [6].

The incubation period ranges from 90–124 days in the wild and from 56–102 days under artificial conditions at temperature of 22–35 °C [5]. In chelonians that exhibit environmentally determined sex females are produced from eggs incubated at temperatures warmer than 29 °C, and males produced from eggs incubated at cooler temperatures [7]. Temperature-dependent sex determination is characteristic of *Testudo hermanni*. A sex ratio of 50% for *T. h. boettgeri* is at threshold temperature 31.5 °C [8].

Literature data on embryonic development in turtles are scant. There are no data on embryonic development of genus *Testudo*. Embryonic development has been described in some soft-shelled turtles (*Pelodiscus sinensis, Apalone spinifera*) [9,10], and marine turtles (*Lepidochelys olivacea*, *Dermochelys sp*.) [11,12], Emydid turtle (*Trachemys scripta*) [13], and in *Chelydra serpentina* [14]. Deformed embryos in reptiles are mentioned in the literature [15,16]. Twins in turtles, crocodiles, lizards, and snakes are rare but reported involving the development of separate individuals of equal or unequal size and varying degrees of malformation known as axis bifurcation [17].

Reproductive success can be influenced by intrinsic and extrinsic factors. Maternal influences have been shown to be a significant intrinsic source of variation in offspring survival and fitness. Females can influence reproductive performance through nest site choice. Extrinsic factors related to nest microhabitats, such as incubation temperature and substrate water potential, influence tortoise hatchings and the survival rate of hatched young tortoises [18]. Hatching can take several hours or even days [19].

We are convinced that a high hatching quality is essential for the further development and welfare of tortoises and a more economical way of breeding. Therefore, we aimed to identify the causes of embryonic mortality in *T. h. boettgeri* during artificial incubation on a large farm in Slovenia. The aims of this study are possible improvements in Hermann’s tortoise hatching and increases in the welfare level of breeding.

## 2. Materials and Methods

The research was performed on a turtle farm where several species are bred, mainly Hermann’s tortoise, subspecies *T. h. boettgeri*. The tortoises are free to roam outdoors and in greenhouses, except hatchlings, which are kept in indoor terrariums. They live indoors for the first three years to protect them from predators. After completion of the laying period, which lasts from May to July, artificial incubation continues for another two months. By the end of August, hatching was complete, and the total hatchery waste of 3557 eggs was examined to determine the causes of mortality.

The hatching artefacts represented 19.2% of all eggs (N = 18,520). The viability rate of incubated eggs was 80.8%. All eggs harvested on the farm were handled, transported and artificially incubated under the same conditions. The eggs taken from the nest marked with a flag were carefully dug out and immediately transported into the pre-hatchery room. In less than six hours eggs were inserted in incubator. The temperature at hatching varied between 31 °C and 32 °C. Humidity was maintained so that the eggs were laid on the substrate (sterile vermiculite), which was occasionally moistened. Average humidity was 80.0% and hatching time 63 days. Incubators were fumigated with F10 SC (Antiseptic solution, pre-diluted in 1:250 concentration, Health and Hygiene, Loughborough, UK) before inserting the eggs.

Candling and auscultation of the embryo’s heart rate and measurement of pulse frequency (Egg Buddy MK1 Digital Egg Heart Monitor for Birds and Reptiles, Avitronics, Truro, UK) were used during incubation.

After the incubation period, all unhatched eggs were analyzed. The eggs were grouped according to fertilization and causes of embryonic mortality. The embryos were classified into three groups according to different size. Tortoises classified in group 1 were a maximum of 1.0 cm in length, group 2 included tortoises ranging in size from 1.0 cm to a maximum of 1.5 cm, and group 3 included tortoises longer than 1.5 cm. In the last group, inability to peck the shell and different sizes of yolk sacks were also observed. The percentages of unfertilized eggs, desiccated and infected eggs, and embryos were determined. Deformed embryos involving twins were included. At the beginning of embryonic development, the differences between unfertilized and fertilized eggs were evident as white spots, and a few days later, the presence of vessels in fertilized eggs was observed. Swabs were taken from unhatched infected eggs and embryos and analyzed. General bacteriological and fungal examinations were performed [20]. Ectoparasites were sent for further determination.

## 3. Results

The percentages are always calculated based on the total number of inviable eggs (N = 3557).

We detected unfertilized eggs in 61.0%, infected eggs (bacterial and fungal infections) in 52.5%, desiccated eggs in 19.1%, and some of them were also mummified (0.42%). Dead and live deformed embryos (0.48%) and twins (0.11%) were detected. Malformed carapace and missing tail, depigmentation, umbilical hernia, open coelom cavity, missing eye or cyclops, and conjoined twins (parapagus, cephalopagus, thoracopagus, and omphalopagus) were the most commonly found. Many of the eggs folded into two or more groups (e.g., moldy and mummified simultaneously). In our study, 0.73% of eggs were unfertilized, desiccated and moldy at the same time. In 24.1% we could not confirm fertilization because changes were too extensive, and fertilization was evident in only 14.9%. The main reason for this was infection with different pathogens. *Pseudomonas aeruginosa*, *Bacillus* sp., *Purpureocillium lilacinum*, and *Escherichia coli* were most frequently confirmed in infected eggs. In some cases, the bacteria and fungi remained undetermined. The eggs were infested with mites in 2.7%.

Embryos up to 1.0 cm (group 1) were found in 2.2%, embryos from 1.0 cm to 1.5 cm (group 2) in 5.4%, and embryos longer than 1.5 cm (group 3) in 7.3%. From this group, some of the embryos were dead shortly before hatching because they were unable to peck the shell (1.3%). Embryos still alive from the group 2 and the group 3 were confirmed in 0.7%. Of these two groups, ¾ of the embryos belonged to the group 3.

## 4. Discussion

*Testudo hermanni* typically have two nests in the season. A third nest is very rarely described in some *T. h. boettgeri* individuals [21,22]. *T. h. boettgeri* females lay an average 4.3 eggs per nest with a maximum of nine eggs [22]. In our climatic environment (extreme southern edge of Julian Alps), the first mating season usually begins in May and the second in August (Figure 1a,b). The main nesting season on the farm for both laying is in May and June without sexual activity. Females laid 1–7 eggs in the first clutch (average 3.6 eggs) and 1–6 eggs in the second clutch (average 3.1 eggs) (Figure 2a,b). Stvarnik, 2020 [19] confirmed statistically significant correlation between tortoise weight and the number of eggs per clutch but no significant correlation between tortoise weight and average egg weight laid was established.

Differences between tortoise weight and the number of eggs per clutch potentially correlate with diet and health status. *T. h. boettgeri* egg size measured by Bertolero et al. (2011) was 27.9 × 37.4 mm, and the eggs weighed 17.1 g. The average size of *T. h. boettgeri* eggs measured by Stvarnik et al., 2017 [20] was 29.9 × 39.5 mm, and the average weight was 20.7 g. Bertolero et al. [22] determined a slightly lower average value than Stvarnik et al. *T. h. boettgeri* eggs measured by Highfield (1996) were larger at 40 × 29 mm but weighed less (12–14 g) [23]. Hailey and Loumbourdis (1988) found that the average egg weight for *T. h. boettgeri* was 17.8 g, ranging from 10.5 g to 23.5 g [24]. In our study in 2020, the average size of *T. h. boettgeri* eggs was less than that measured in 2017 [4]. The results of the average measurements were 27.9 × 35.6 mm for width and length, and the average weight was 16.5 g (Figure 3a).

During the collection period and throughout the hatching process, the eggs must remain in the same position in which they were collected. There are large differences in the collection of poultry and tortoise eggs. Within a few hours after oviposition, reptile embryos rise to the top of the egg and start adhering to the inner membrane of the shell. If the egg is turned after the embryo has attached itself, the weight of the yolk could impede normal development or tear both the vitelline and extra-embryonic membranes, leading to death or malformations [3]. One of the reasons why reptile eggs are sensitive to movement around their horizontal axis during development is the absence of chalaza [2]. Aubert et al., 2015 [3] found that post-birth mortality in snakes was significantly higher in turned compared to unturned embryos. They suggest that eggs should not be moved from their natural position. Tortoise egg collections are very important part of a successful hatching (Figure 3b). Within six hours, the eggs are inserted in the disinfected incubators. Antiseptic solution F10 SC is used for fumigation (Figure 4).

During incubation eggs can be examined with candling [25,26] (Figure 5a). The white of a fresh egg is cloudy and very thick (Figure 5b). As the egg ages, the white becomes almost transparent and thin as air passes through the pores in the eggshell (Figure 5c). Each yolk is covered with a thin transparent membrane that prevents the yolk from breaking. This membrane becomes increasingly thinner as an egg ages, so fresh yolks stand taller and are less likely to break. As an egg ages, the yolk absorbs water from the albumen. The evaluation of the basic composition, amino acid and fatty acid profiles and the presence of certain trace elements in *T. h. boettgeri* eggs has already been mentioned by Stvarnik et al. [4].

The eggs were candled with light to assess whether embryonic development is present. Desiccated eggs were also confirmed with candling (Figure 6a–e).

The development egg monitor (Egg Buddy MK1 Digital Egg Heart Monitor for Birds and Reptiles, Avitronics, Truro, UK) is used by many breeders to identify embryonic vitality in eggs of reptiles and birds [26,27] and is also suggested for use in small vertebrates under 40 g [28]. Until this study, heart monitoring was not regularly used on the farm (Figure 7). However, examination of the heartbeat may help in the determination of embryonic deaths.

The percentage of egg failure represented in our farm was 19.2% of all eggs. There are a lack of literature data about egg failure of *T. h. boettgeri* so we cannot compare our results with others. There are more data about this species under natural conditions [5,22,29]. Artificial incubation in farm breeding of *T. h. boettgeri* is described by Eendebak, 2001 [8]. He presented egg viability at different temperatures. At hatching temperature 31 °C mortality was 25%, and at 32 °C, he found 20% mortality. We found 19.2% mortality at temperature from 31 °C to 32 °C.

Humidity and temperature are important factors for proper embryonic development and hatching. At high levels of desiccation, embryonic mortality increases. Mortality appears later in incubation when water stores decline below some critical value. The limiting value is likely one that compromises either tissue osmotic concentration and/or potentially blood flow in the cardiovascular system. Analyses of embryo and yolk mass indicate that desiccation results in a decreased ability to use yolk to produce tissue mass [26]. In our study desiccated eggs in various stages were confirmed in 19.1% (Figure 8a,b).

The reason could be due to cracked eggs and poor egg manipulation. The improper nutrition of the breeding parents and reproductive diseases of the females (e.g., inflammation of the oviduct) can also be the reason (Figure 9a,b). Cracked or thinner eggshell were noted at *T. h. boettgeri* (Figure 9c). Some previous studies have shown that artificial incubation of small eggs from *Testudo hermanni* is not successful [18,19]. Based on this fact, we changed the protocol for *T. h. boettgeri* in our farm. Eggs weighing less than 10 g are no longer hatched. The possible reason could be faster desiccation of small eggs. Mummification of tortoise embryos has not been described as a problem during incubation, but it seems to be a very important fact in our study. In the future, it will be necessary to carefully study the factors affecting humidity and resulting drying of eggs/embryos on the farm (Figure 10a–j). In our study, 0.73% of eggs were unfertilized, desiccated, and moldy at the same time (Figure 11a–c). Because of the previously mentioned reasons, the occurrence of desiccated eggs indicates obligatory control of humidity and temperature in the incubator.

Important critical points for successful hatching are also weight of eggs, their shape and thickness of shell. Typically, individual tortoises give eggs of similar size value or shape. Differences may be due to unbalanced meals and/or health problems (e.g., metabolic diseases, ovarian or oviduct infections, other reproductive difficulties, problems in young hatching tortoises). Differences also depend on the species [7]. Mechanical damage or cracking of the shell can happen at the time of tortoise laying (Figure 12).

Among the embryos at different sizes, we found desiccated and/or moldy embryos, infected embryos, deformed embryos including twins, and also live embryos. At the beginning of development, when the vessels were not always seen or when the eggs were infected and the changes were too extensive the estimating of fertilization could be unreliable (Figure 13a,b). Occasionally, during ovulation a blood vessel in the stroma ruptures and bleeds onto the surface of the follicle. The blood spot remains even if the embryo dies before obvious development of embryonic blood islands. When the egg is opened at the end of incubation, ovarian blood spots may be confused with embryonic blood islands [7]. In these mentioned cases, we could not accurately determine fertility of eggs. When we look for the presence of an embryo, it helps to look at the surface of the yolk sack on which the embryo floats in all stages of development.

There is a lack of literature data about the reasons for egg failure in *Testudo hermanni* and also in other turtle species, especially regarding pathogen determinations (bacterial, fungal, and/or parasitic infections/infestations). Sampling must be planned in the future to improve incubation conditions and prevent the spread of disease (Figure 14a–c). Many infections are also found in the breeding parents, which have to be treated before the hatching season. Some of the eggs were so infected that gas was formed in the contents. These eggs exploded on rough manipulation or just on touch (12.1%). This finding is not a good indicator for the transfer of pathogens during incubation, so it is necessary to regularly dispose infested eggs and periodic candling is recommended.

Bacterial and fungal infections were confirmed in 52.5%. *Pseudomonas aeruginosa*, *Bacillus* sp., *Purpureocillium lilacinum*, and *Escherichia coli* were frequently confirmed in infected eggs. (Figure 15a–c). Tortoises with oedema of the neck and other parts of the body with infected yolk sacks were often diagnosed. A thickened wall of the yolk sack was observed. Many were still alive but died in a few hours. The content was changed, and a characteristic odor was present. *Escherichia coli* was isolated in these cases (Figure 15d,e).

Infestation of eggs/embryos has not been described as a problem during incubation. However, similar to desiccation, it seems to be a very important fact. In our study, 2.7% of the eggs were infested with mites that completely destroyed the eggs. It looks like a powder that moves. The powder produced by astigmatic mites on their substrate contained more than 50% mites. The proliferation of detritivorous mites (Astigmata) in eggs is quite typical of an environment with a lot of organic mass (the contents of the eggs). A Mesostigmata mites are strictly predators and feed on mites and other invertebrates. The presence of a predator in such a context is quite common [30] (Figure 16a–e).

There are no data about development, deformed embryos and twins for *Testudo* species. In our study, dead and alive deformed embryos were detected in 0.48% (Figure 17a–h) and twins in 0.11% (cephalopagus, parapagus, thoracopagus, omphalopagus) (Figure 18a,b).

Among deformed embryos, severe deformation of the shell, umbilical hernia, and depigmentation were found. Tortoises without eyes, nares and mouth, tortoises with one eye (cyclops) also occurred, and an infected yolk sack was present in most of them. We noted depigmentation of skin and shell in all deformed embryos except in twins. Malformed carapaces and missing tails were found in 0.92% hatched tortoises. These babies were not a part of egg failure study but are still important for hatching conditions improvement. They hatched by themselves and mostly died in a few months. In some cases, the shells were covered with mold (Figure 19a,b).

By examining the unhatched eggs, we discovered an additional 0.67% live embryos (24 live turtles). Seventy-five percent were longer than 1.5 cm (group 3). These embryos would have a chance to survive if we provided proper care for healthy babies. Among them were also deformed embryos and malformed twins. In the future, a systematic study of the developmental stages of genus *Testudo* has to be carried out (Figure 20a–h).

The main threats to the survival of Hermann’s tortoise (*Testudo hermanni*) are habitat loss due to agricultural expansion and intensification, agrochemicals and other pollution, urbanization and development of tourism infrastructure, forest fires, genetic pollution, road mortality, use of alternative medicine and collection for the pet trade [29,31,32]. Despite the endangered status in their natural habitats, Hermann’s tortoise remains one of the most popular tortoises to keep as pets because they are small and have a long lifespan of over eighty years [33]. Given the reasons mentioned above, especially the pet trade [34], artificial breeding is an important component for conservation in the wild. People can buy pets from breeding farms and no longer hunt tortoises from the wild. Proper hatching is essential for tortoise welfare and allows a more economical method of breeding

## 5. Conclusions

To increase the percentage of hatching and improve the welfare of tortoises, it is necessary to monitor some parameters and make some changes. It is necessary to regularly monitor total egg failure and improve the conditions for hatching to obtain healthy young tortoises. Eggs must be examined before hatching, and all small eggs (less than 10 g), eggs with different shapes, and those with broken shells must be eliminated. To reduce infections, eggs should be candled before and periodically during artificial incubation. All unfertilized and dead embryos must be removed immediately. Fumigation with disinfectants prior to incubation is required. The following critical points are important to establish and improve proper temperature in conjunction with humidity, incubation of “clean” eggs on sterile substrate, control for the presence of mites, and a healthy parent group of tortoises. Monitoring of the parent tortoises is also necessary. Proper hatching will keep many tortoises alive that might otherwise die in infancy or suffer from many health problems throughout their lives.

## Figures and Tables

**Figure 1 animals-11-00478-f001:**
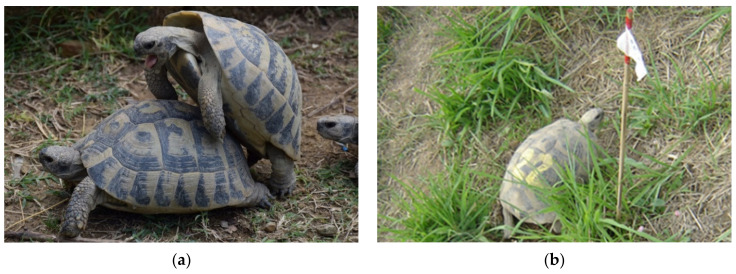
Mating season. (**a**) The mating season for *Testudo hermanni boettgeri* typically lasts from April to mid-July and depends on climatic conditions. (**b**) The breeder marks the nest with a flag to make it easier to collect eggs.

**Figure 2 animals-11-00478-f002:**
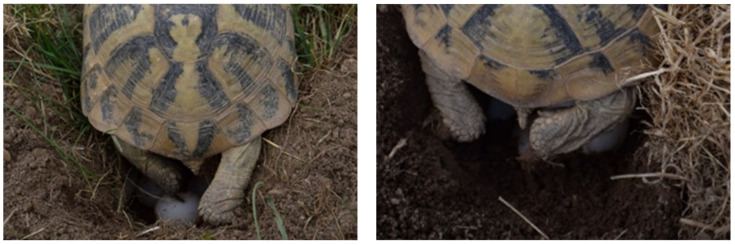
Digging, arranging a nest, and laying eggs. (**a**) (left column) The female laid eggs in a dug hole in the soil. Each egg aligned itself with its claws at the exact location in the nest. This female laid three eggs. (**b**) (right column) This female laid five eggs and covered them with soil after the last egg was laid. Alignment may take several minutes for a single egg.

**Figure 3 animals-11-00478-f003:**
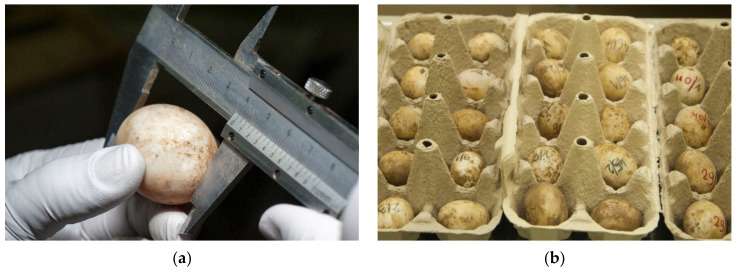
Tortoise egg collections. (**a**) For the research study, the eggs were measured with Vernier caliper before being placed in the incubator. Sterile gloves were used. (**b**) Tortoise eggs are collected directly from the hole the tortoise has dug. On the surface of the shells, there is still the soil that remains on them during the whole hatching process. Eggs for research purposes were marked for traceability.

**Figure 4 animals-11-00478-f004:**
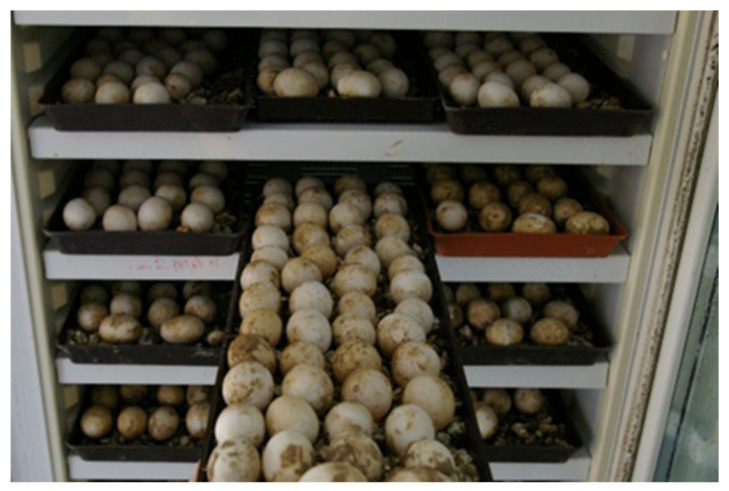
Insertion of tortoise eggs into the incubator. In total, 18,520 eggs laid by *Testudo hermanni boettgeri* were inserted into the incubator. The viability rate of incubated eggs was 80.8%.

**Figure 5 animals-11-00478-f005:**
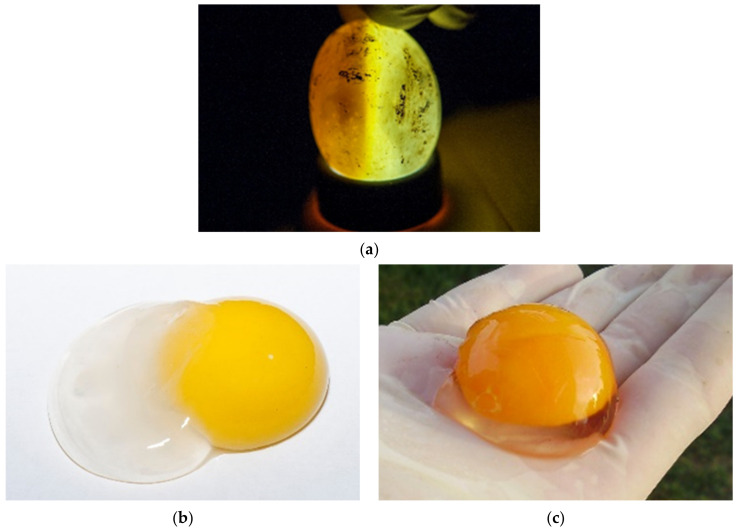
Examination of eggs during artificial incubation—unfertilized eggs. (**a**) In this case, an unfertilized egg was observed. When the light shone through the egg, considerable empty space was noted. It appeared that embryonic development had not started. (**b**) After we cracked this egg, the result was the same. We also saw that chalaza was not present. This egg was incubated a couple of weeks before we examined it and was not fresh. The albumen and the structure of the yolk are observed in this image. (**c**) In this figure, there is a fresh egg one hour after hatching. The structure is different from the non-fresh egg in Figure 5b.

**Figure 6 animals-11-00478-f006:**
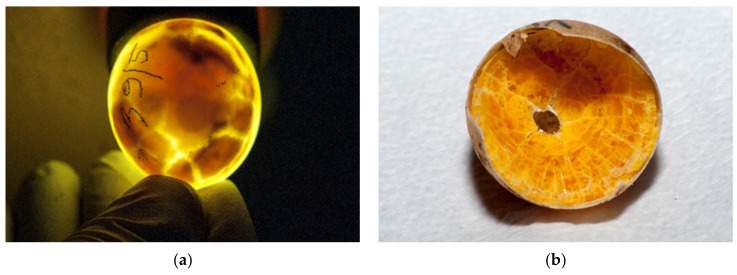
Examination of eggs during artificial incubation—desiccated eggs. (**a**) In this egg, desiccation was seen with a candling lamp. (**b**) After we cracked this egg, the desiccating yolk of the unfertilized egg was found. (**c**) In the late stage of development, movement of the embryo can be observed. In this figure, the shadow is the dead embryo, and yolk is visible bellow the embryo. The vascularized grid was not visible. (**d**) After the egg was broken, a dead, desiccated embryo was seen. Vascularization was observed. (**e**) Another unfertilized egg where the drying process started.

**Figure 7 animals-11-00478-f007:**
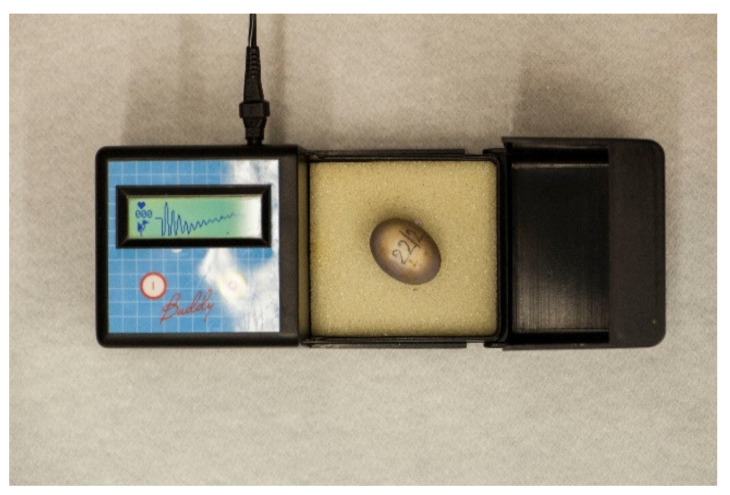
Egg viability can be determined with candling, or by auscultation of the embryo’s heart.

**Figure 8 animals-11-00478-f008:**
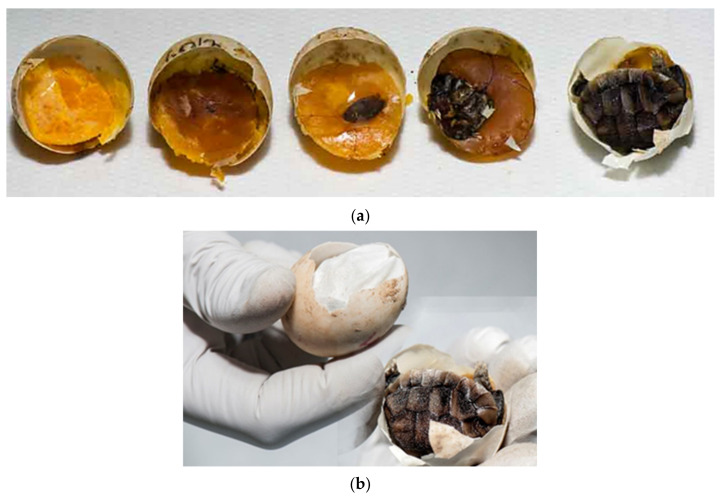
Examination of eggs after artificial incubation—desiccated eggs. (**a**) Both unfertilized eggs and fertilized eggs with embryonic development at different stages may be desiccated. Figure presents (from left to right) unfertilized eggs and embryos in different sizes: embryos up to 1.0 cm (group 1) (two embryos), embryos from 1.0 cm to 1.5 cm (group 2), and embryos longer than 1.5 cm (group 3). (**b**) The percentage of moisture is important throughout the hatching period. Sometimes dehydration does not occur until in the last few days before hatching.

**Figure 9 animals-11-00478-f009:**
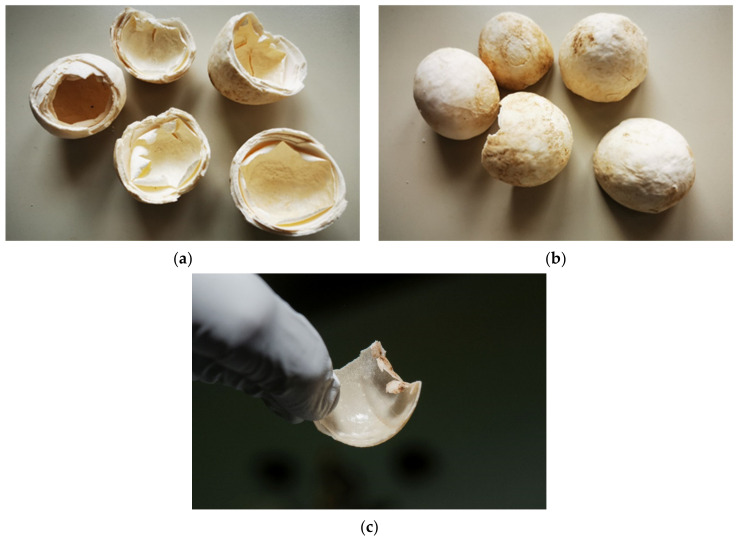
Shell abnormalities. (**a**,**b**) Shell deformation of *Chelonoidis carbonaria* was present in all eggs in the nest. (**c**) Thinner eggshell of *Testudo hermanni boettgeri*.

**Figure 10 animals-11-00478-f010:**
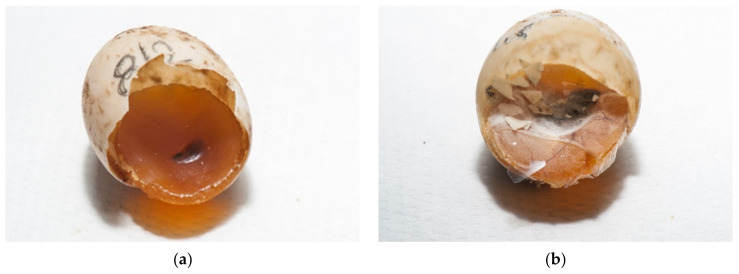
From desiccation to mummification. Figures show embryos in different sizes. (**a**) Embryos up to 1.0 cm (group 1); (**b**) embryos from 1.0 cm to 1.5 cm (group 2). (**c**–**j**) Embryos longer than 1.5 cm (group 3). We can also see varying degrees of dehydration: from mild to complete, where we talk about mummification (**a**,**b**: mild desiccation; **c**,**d**: moderate desiccation; **e**–**j**: mummificated embryo; mummificated embryo). We often detected the presence of mold in such embryos (**g**–**j**: mold is seen).

**Figure 11 animals-11-00478-f011:**
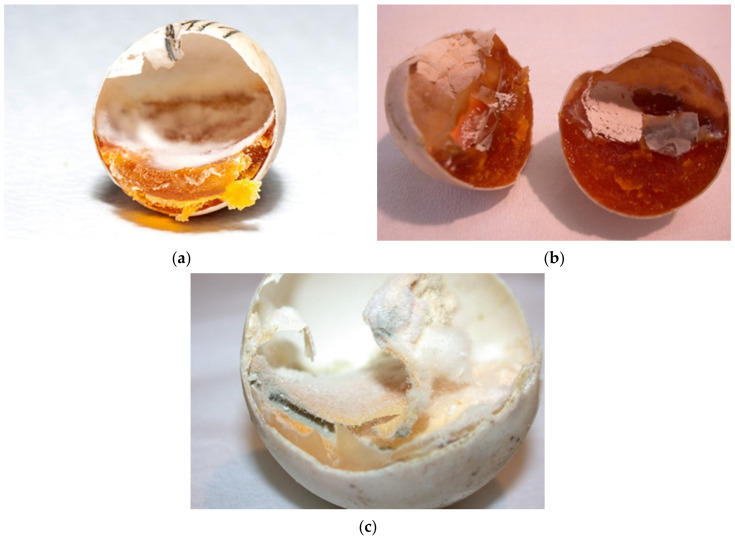
Moldy eggs. (**a**–**c**) Mild desiccation is shown in the first two figures, and moderate desiccation in the third figure.

**Figure 12 animals-11-00478-f012:**
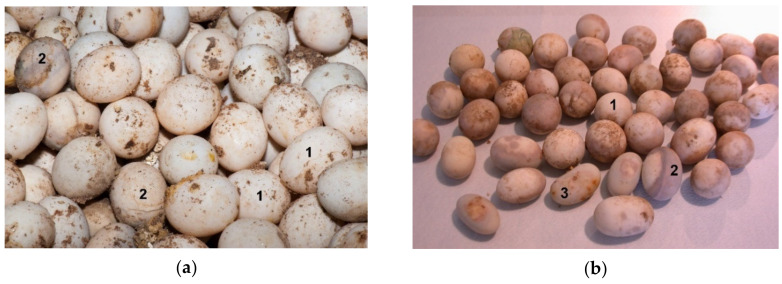
Examination of unhatched eggs after artificial incubation. (**a**) Soil and hatching substrate (vermiculite) are still present on the surface of eggs. Some eggshells are broken. The shell may be completely white (mostly unfertilized eggs) or have unevenly distributed darker spots, freckles, or entire areas under the shell (eggs in various development stages, infected eggs): 1 unfertilized egg, 2 infected eggs in various development stages. (**b**) Different size (weight) and shape of eggs: 1 round; 2 ovals, 3 irregular. (**c**) Cracked eggs of *Astrochelys radiata* and *Centrochelys sulcata* (on the figure) were glued with beeswax and propolis to prevent desiccation and infection of the embryo. Eggs of different species (*Astrochelys radiata, Centrochelys sulcata*, *Chelonoidis carbonaria, Malacochersus tornieri, Indotestudo elongata, Indotestudo forstenii*) are shown in Figure 12b,c.

**Figure 13 animals-11-00478-f013:**
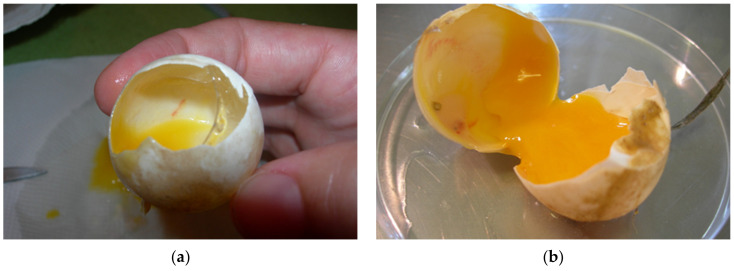
Fertilized eggs. (**a**,**b**) Vessels are seen but with egg candling we determined them as unfertilized.

**Figure 14 animals-11-00478-f014:**
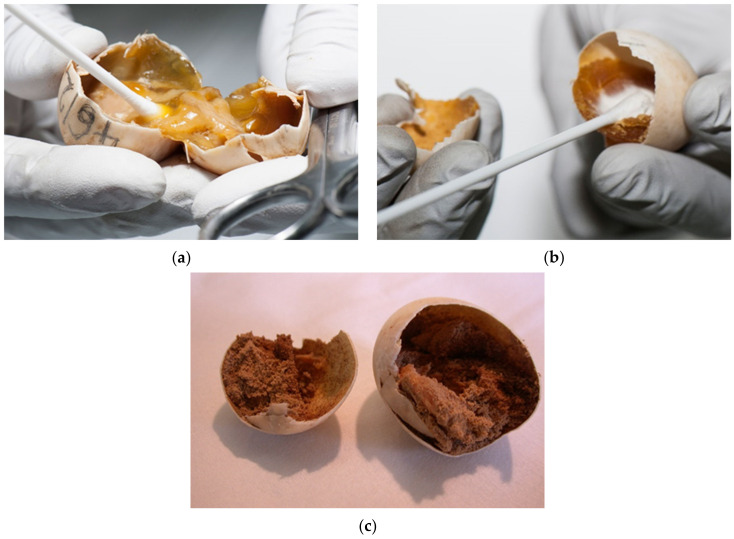
Sampling for laboratory testing. (**a**) The sample was taken because we suspected a mixed bacterial infection. (**b**) The sample was taken because we suspected a mold infection. (**c**) We sampled the eggs in which the contents of egg were changed into a powder. In these cases, we suspected mites.

**Figure 15 animals-11-00478-f015:**
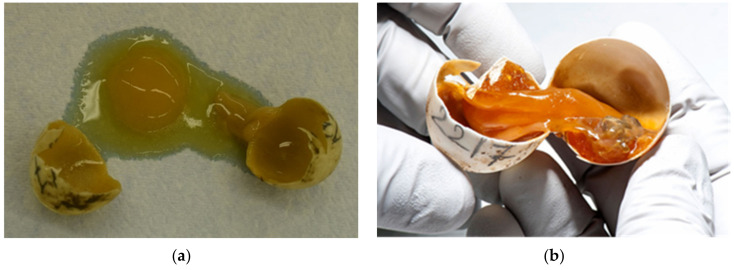
Infestation in different development stages. (**a**) This egg was unfertilized and infected at the same time; (**b**) Infected embryo from 1.0 cm to 1.5 cm (group 2) in which yolk content was desiccated. (**c**) Completely destroyed egg, autolysis due to post-mortem changes and unknown pathogens were found. (**d**,**e**) Tortoise with oedema of the neck and other parts of the body with infected yolk sacks.

**Figure 16 animals-11-00478-f016:**
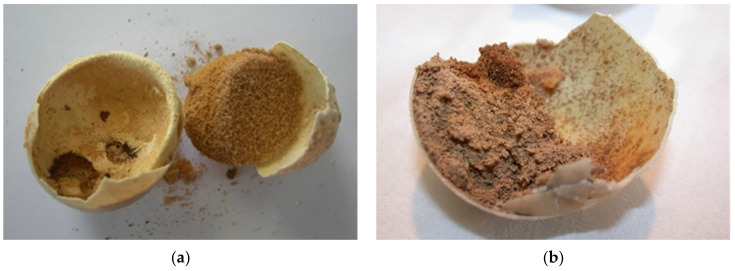
Infestation with mites. (**a**,**b**) Infestation of eggs causing destruction of egg contents. (**c**) An adult female with egg. (**d**) Two adults (male and female) and a larva. (**e**) A Mesostigmata mites. Images (**c**–**e**) are obtained at 100× magnification.

**Figure 17 animals-11-00478-f017:**
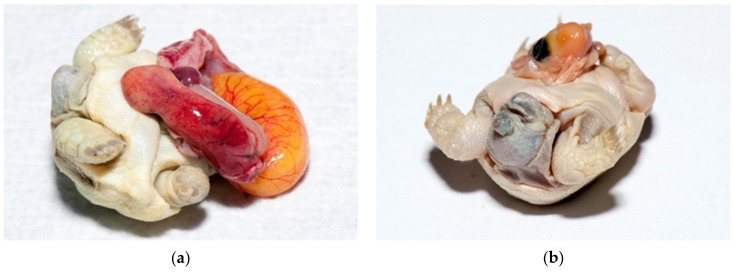
Dead deformed embryos. (**a**) Umbilical hernia was diagnosed on this embryo. The liver, small intestine with pancreas and heart were seen outside of the celom cavity. Blood vessels were well visible on yolk sack. Pigmentation of skin and shell was poor. Tortoise was still alive. (**b**) Tortoise with one eye (cyclop), with hernia and poor pigmentation died at the late development stage. (**c**) Two tortoises without eyes, nares and mouth; one tortoise also had an infected yolk sack (at the right). Both were still alive. (**d**); Two tortoises with infection. Infection and oedema were present subcutaneously. The left tortoise is normally pigmented, and the right tortoise has depigmented shell. (**e**) Deformation was observed on face. Tortoise lived only a few days. (**f**) Severe deformations were seen on face. Tortoise with no eyes and nares. Depigmentation is present. The tortoise was dead. (**g**) Severe deformation of the shell is also quite common. (**h**) This live tortoise, with slight depigmentation, lived two months. When we found a viable tortoise at examination of unhatched eggs, we placed it on clean moist substrate and warm surroundings. In this case, warm water was in the bottle. Then, all healthy tortoises were transferred to an incubator where first aid was provided.

**Figure 18 animals-11-00478-f018:**
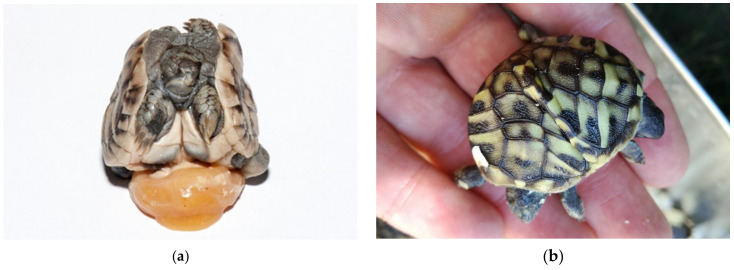
Twins. (**a**) Cephalopagus—tortoise with eight legs and one head was found. An infected yolk sack was also noticed. This tortoise had one heart, two livers, two stomachs and intestines and four kidneys. The gender was unknown. It died after two days. (**b**) Parapagus—tortoise with two heads and six legs was found. This tortoise had two hearts, two livers, and two stomachs but only one intestine and two kidneys. This male was alive for 18 months.

**Figure 19 animals-11-00478-f019:**
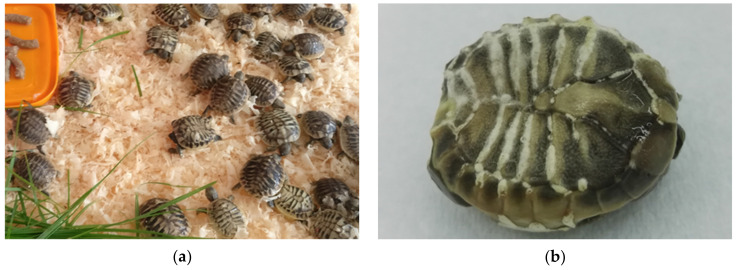
(**a**,**b**) Hatched tortoises with malformed carapaces and missing tails. In these cases, the shells were often covered with mold.

**Figure 20 animals-11-00478-f020:**
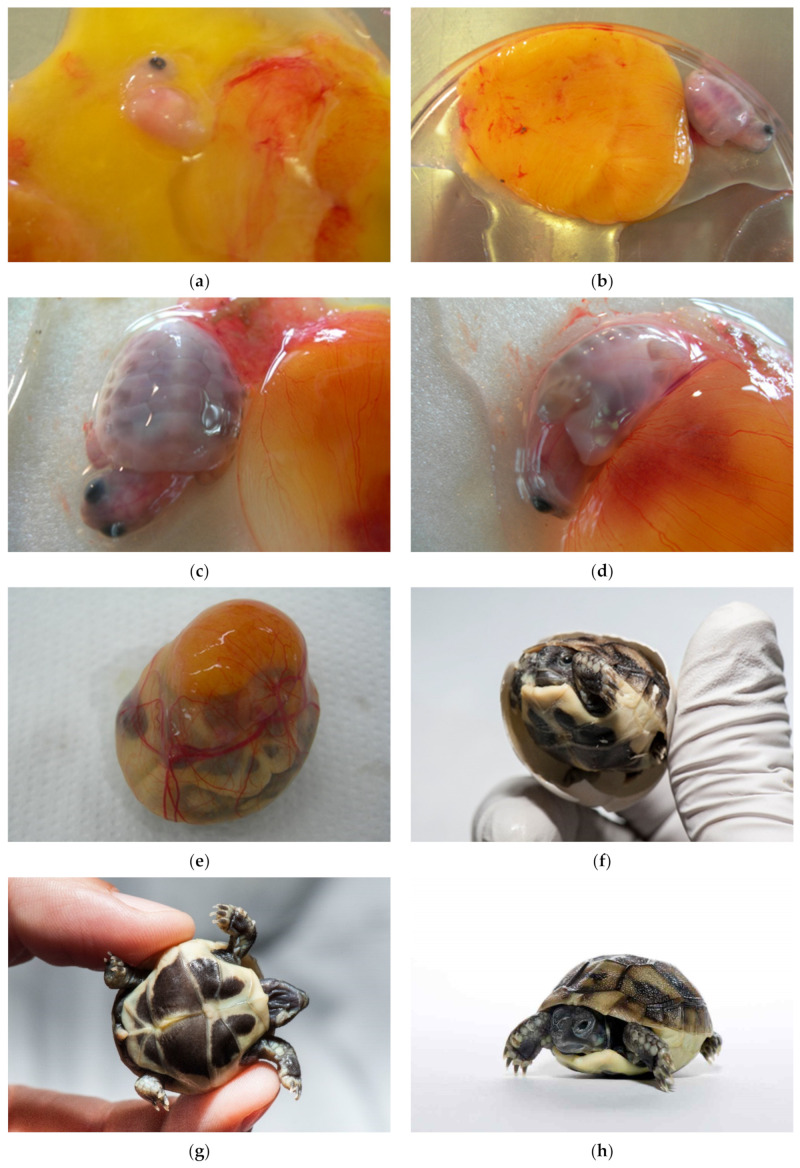
Alive. (**a**) Alive embryo up to 1.0 cm (group 1). (**b**,**c**) Alive embryos from 1.0 cm to 1.5 cm (group 2). All embryos size up to 1.5 cm (group 1 and 2) died within a few hours. (**d**) Alive embryo floating on the yolk sack. Vessels are clearly visible. (**e**–**h**) This newly laid tortoise was successfully pecked through the eggshell. It is still alive.

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
