# Peer review of "Monitoring of Unhatched Eggs in Hermann’s Tortoise (Testudo hermanni) after Artificial Incubation and Possible Improvements in Hatching"

_animals, 2021, doi:10.3390/ani11020478_

Round 1
Reviewer 1 Report
1. line 31 In abstract need to say how many total eggs observed: 3,557 of 14,963 (?) to make clear how many total used to equal the total percentages studied.
2. This paper has some useful information but it has no form that can be easily used by the reader. See for example the elephant article just published with Intro., Methods and Materials, Results, Discussion and Conclusion. The data and discussion are mostly difficult to locate in the legends for the figures.
3. The figures are mostly useful but results and discussions should be moved to appropriate major paper headings.
4. The term "infertility" has been a major issue in turtle egg evaluation. Other authors have found that often the infertility diagnosis is actually a very early embryonic death. I believe the authors see this in their candle technique which does not show the white spot but does show the small faint blood vessels. The Miller et al. references show and discuss this.
line 39- "creatures" and throughout paper is not a scientific term. Possibly substitutions "deformed" or malformed" embryos. Also for rotten eggs "addled" eggs is often used.
line 61- Chalaza is nice term- more complete definition would be useful since this topic is very important in reptiles.
line 117- suggest deformed embryos and not "creatures"
line 126 - suggest "number of inviable eggs" more appropriate
line 153 for both "clothing" I do not understand this term and its meaning.
line 215 to assess (not assessing)
line 222 "this system" Does this refer to Egg Buddy MK1? Not clear since it is first noted in Fig. 7 legend.
line 233 "time of tortoise (laying)" better than "layed".
line 237 use 14,963 not "14.963"
line 238 "unbalanced meals and/or ..." this speculation is out of place in a legend. Possibly for a "Discussion"
line 241 references to 5b and 5c seem incorrect.
Figure 8 - What do numbers on the photos refer to?
Figure 10 photos suggest early embryonic death, so the were actually fertile and should not be included as "infertile".
line 275 "fallopian tube" not use in reptiles better word is oviduct.
line 293 signs not "sings"
line 319 with ?? not "wah"
line 321-322 warm not "worm"
line 321 In not "It"
line 324 "egg failure" suggest egg failure study
line 389 the Miller et al. reference is very important.
Author Response
Dear REVIEWER 1!
Corrections are made according to your suggestions as they follow:
1 line 31 In abstract need to say how many total eggs observed: 3,557 of 14,963 (?) to make clear how many total used to equal the total percentages studied.
The number of total eggs observed has been added.
- This paper has some useful information but it has no form that can be easily used by the reader. See for example the elephant article just published with Intro., Methods and Materials, Results, Discussion and Conclusion. The data and discussion are mostly difficult to locate in the legends for the figures.
Form has been improved.
- The figures are mostly useful but results and discussions should be moved to appropriate major paper headings.
Data from the legends for the photos were moved to appropriate major paper headings.
- The term "infertility" has been a major issue in turtle egg evaluation. Other authors have found that often the infertility diagnosis is actually a very early embryonic death. I believe the authors see this in their candle technique which does not show the white spot but does show the small faint blood vessels. The Miller et al. references show and discuss this.
Findings of Miller and Dinkelacker (2008) in chapter Reproductive structures and strategies of turtles in the Biology of turtles has been added in “Discussion”.
line 39- "creatures" and throughout paper is not a scientific term. Possibly substitutions "deformed" or malformed" embryos. Also for rotten eggs "addled" eggs is often used.
We changed the term to deformed embryos and malformed twins.
line 61- Chalaza is nice term- more complete definition would be useful since this topic is very important in reptiles.
Aubert et all (2015) reference has been added but as a fresh egg was not the object of our investigation we believe that more explanation is not necessary. We have stressed some sentences in the text as well.
line 117 suggest “deformed embryos “and not “creatures” taken into account and changed
line 126 suggest "number of inviable eggs" more appropriate taken into account and changed
line 153 for both "clothing" I do not understand this term and its meaning.
“Clothing” was replaced by “laying”.
line 215 to assess (not assessing) taken into account and changed
line 222 "this system" Does this refer to Egg Buddy MK1? Yes
The term “this system” was replaced by “heart monitoring”.
line 233 "time of tortoise (laying)" better than "layed". taken into account and changed
line 237 use 14,963 not "14.963" taken into account and changed
line 238 "unbalanced meals and/or ..." this speculation is out of place in a legend. Possibly for a "Discussion" taken into account and moved to discussion
line 241 references to 5b and 5c seem incorrect.
There was our mistake, they refer to the picture 8b and 8c instead of 5b and 5c.
Figure 8 - What do numbers on the photos refer to?
Explanation of the numbers is under the photos
Figure 10 photos suggest early embryonic death, so they were actually fertile and should not be included as "infertile"
Our mistake, taken into account and deleted
line 275 "fallopian tube" not use in reptiles better word is oviduct. taken into account and changed
line 293 signs not "sings" This sentence was deleted.
line 319 with ?? not "wah" taken into account and changed
line 321-322 warm not "worm" taken into account and changed
line 321 In not "It" taken into account and changed
line 324 "egg failure" suggest egg failure study taken into account and changed
line 389 the Miller et al. reference is very important.
Reference has been added but we could not find any correlation due to the authors description of the histology of marine turtle Dermochelys.
We have also added Yntema et all (1968) in references but they are also describing a water turtle. As there are no data about histology of Testudo hermanni we may decide to perform that in the future.
Thank you for your suggestions.
Sincerely yours author and coauthors.
Alenka DovÄŤ

Reviewer 2 Report
This is an impressive study, analysing the causes of incubation failure in a large number of eggs collected from the endangered tortoise, Testudo hermanni. Populations in the wild are threatened by a variety of processes, including, primarily, loss of suitable habitat. Turtle farms, such as the one in this study, can be an important source of healthy individuals for return to the wild, as well as catering for the large pet trade. A total of 3,557 ‘waste’ eggs were examined from a total of 18,520 and 61% of these were found to be unfertilised. Hatching success from these figures is an impressive 80.8%, attesting to the effective husbandry measures already in place in the farm. Of the failed eggs, 52.5% had bacterial and fungal infections, 19.1% were desiccated and 0.42% mummified. There was also a small percentage of anatomical malformations, ~ca 0.5%. The authors are to be congratulated on their careful analysis of the 3,557 failures and have identified ways by which these losses can be reduced. Primarily amongst these is the control of infection. They do not say whether the vermiculate, on which the eggs are incubated, is sterilised or not. If not, it should be. Controlling the humidity in the incubators is also essential if loss from desiccation and mummification of eggs is to be avoided. In Figure 2b an egg is being held in the hand and measured with a micrometer. What is not clear is whether the operator is wearing sterile gloves. If not, this would be an obvious source of potential infection and needs to be implemented whenever eggs are touched. The study also shows that very small eggs (<10g) need to be discarded as they have a very poor chance of developing. Candling also needs to be used routinely and unfertilised eggs removed as this is the largest cause of incubation failure. The paper is well written, the English clear and, although the many figures are informative, there are some that could be deleted if space is a problem for the journal (e.g. Figs 1, 4, 10, 12, 14, and 16).
Author Response
Dear REVIEWER 2!
Corrections are made according to your suggestions as they follow:
-This is an impressive study, analyzing the causes of incubation failure in a large number of eggs collected from the endangered tortoise, Testudo hermanni. Populations in the wild are threatened by a variety of processes, including, primarily, loss of suitable habitat. Turtle farms, such as the one in this study, can be an important source of healthy individuals for return to the wild, as well as catering for the large pet trade. A total of 3,557 ‘waste’ eggs were examined from a total of 18,520 and 61% of these were found to be unfertilised. Hatching success from these figures is an impressive 80.8%, attesting to the effective husbandry measures already in place in the farm. Of the failed eggs, 52.5% had bacterial and fungal infections, 19.1% were desiccated and 0.42% mummified. There was also a small percentage of anatomical malformations, ~ca 0.5%. The authors are to be congratulated on their careful analysis of the 3,557 failures and have identified ways by which these losses can be reduced. Primarily amongst these is the control of infection.
They do not say whether the vermiculate, on which the eggs are incubated, is sterilised or not. If not, it should be.
Prior inserting the eggs incubator is fumigated with F10 SC antiseptic solution. We have stressed it in the text as well.
Controlling the humidity in the incubators is also essential if loss from desiccation and mummification of eggs is to be avoided. In Figure 2b an egg is being held in the hand and measured with a micrometer. What is not clear is whether the operator is wearing sterile gloves. If not, this would be an obvious source of potential infection and needs to be implemented whenever eggs are touched.
The use of gloves is seen in the figure. We have stressed it in the text as well.
The study also shows that very small eggs (<10g) need to be discarded as they have a very poor chance of developing. Candling also needs to be used routinely and unfertilised eggs removed as this is the largest cause of incubation failure. The paper is well written, the English clear and, although the many figures are informative, there are some that could be deleted if space is a problem for the journal (e.g. Figs 1, 4, 10, 12, 14, and 16).
In one of our mails to the editor we have asked about the number of figures we intend to add and it was noted not to be a problem for the journal.
Thank you for your suggestions.
Sincerely yours author and coauthors.
Alenka DovÄŤ

Reviewer 3 Report
I have read the manuscript by Alenka DovÄŤ and colleagues, submitted to the journal Animals. The manuscript describes the causes of embryonic mortality during incubation in the tortoise Testudo hermanni. The study is based on large sample (more than 3500 failure eggs), from the artificial incubation. Data on embryonic mortality in turtles and tortoises are scarce, thus the data presented in the manuscript are worth to be published. However, I think the version of the manuscript should be improved. I hope that the following comments can be used to improve the manuscript.
General comments:
I feel that the manuscript could be arranged in a better way.
For example, the Introduction section:
-- the Authors start (the 1st paragraph of the manuscript) from “The main threats to the survival of Hermann’s tortoise…”, but basic information about the species are presented in the second paragraph (“TH is a medium-sized terrestrial species that occurs naturally in the European Mediterranean. (…)”.
-- Information that about the yolk content in reptiles, and that reptile eggs do not contain chalaza, are important for the study. However, in the version of the text, such information are in the 3rd paragraph of the Introduction, just between information about size and distribution of the species (paragraph 2.), and information about the tortoise ecology/activity (paragraph 4.).
It was difficult, for me, to follow the Discussion section. The text of the section is divided by photos. Additionally, in the captions of the photos, there are a lot information about results(?) For example: lines 169-170, from caption of the Figure 3 “The weight of the tortoise and its average egg weight were compared. No statistically significant correlation was between tortoise weight and average egg weight laid (r = 0.124, p = 0.459).” Forgive me for my possible misunderstandings, but I think that results from other studies (i.e., from literature), and results from this study, are mixed in the figure captions; it could be confusing for readers. [By the way: the mentioned results of the statistical analyses is incomplete: there is no information on the sample size. There is no information about the statistical analyses in the Materials and methods section, either.]
Photos are really good. However, in such version, they are rather good for ‘supplementary materials’, I think. In the text, the photos/figures are cited only once: line 125 “The egg failures and the causes of embryonic mortality are shown in Figures 1 to 19.” Thus, or the figures are not essential for the text (then can be publish as ‘supplementary materials’), or the text and figures in the Results and Discussion sections should be arranged in an another way.
To recapitulate: I recommend to try to arrange the text in a better way.
Additionally, in the Discussion section I would like to see more important for the study information, for example: if such proportion of egg failure is typical for tortoises? // or typical for the studied species? // for other reptiles? Or, maybe, there is lack of such data for tortoises? Another important question: if the results are typical for eggs from farms, but for ‘wild’ populations situation is different?
To say it short: I am not expert in the area (I believe, Authors are such experts), and without good discussion I have problems with evaluating significance of the collected data.
Several specific comments
I am not sure, if using the abbreviations “TH”, “THH”, “THB”, is good for the text… However (if Authors will decide to leave the abbreviations), I recommend not to use such abbreviations in the figure and table captions. Typically, in scientific papers, tables and figures should have self explanatory captions, i.e., captions should provide sufficient information to the readers to understand the table/figure without looking for information in the text. Thus, name of the studied species is crucial in any figure/table caption.
The Table 1 is difficult to understand, and should be arranged in a better way.
lines 76-79. I am not sure why just the literature position have been cited here: there are cited papers on soft-shelled turtles, marine turtles, Emydid turtles… However, for example, one of the the most classical paper in the subject is not mentioned (Yntema C. L., 1968. A Series of Stages the Embryonic Development of Chelydra serpentina. J. Morph. 125: 219-252).
I understand that is no possible (and no necessary!) to cite all papers, but in the version it looks like “a short literature review” in the subject, i.e., on embryonic development. But what about tortoises? The study concerns on Testudo species. If there are no data on embryonic development of tortoises – write down it.
Abstract:
-- results presented in the abstract are not easy to read,
-- there is lack of summary of the study, in the Abstract.
I found no sufficient information on methodology of the study, for example, on procedures during handling and transporting eggs. It is important for the study, as e.g., “In the case of incorrect movement or rotation of the egg, the embryonic vessels may rupture and consequently cause embryonic death [7].” lines 61-63.
lines 106-107: “The temperature at hatching varied between 31 and 32 °C.” Why such temperature were used? As “Temperature-dependent sex determination is characteristic of TH” (lines 72-73), I am not sure if the used incubation conditions are the best ones.
lines 112-115: “The embryos were classified into three groups of embryonic development. (…) a maximum of 1.0 cm in length, (…) in size from 1.0 cm to a maximum of 1.5 cm, and (…) longer than 1.5 cm.”
thus: “embryonic development” or just “embryos of different sizes”(?)
Author Response
Dear REVIEWER 3!
Corrections are made according to your suggestions as they follow:
I have read the manuscript by Alenka DovÄŤ and colleagues, submitted to the journal Animals. The manuscript describes the causes of embryonic mortality during incubation in the tortoise Testudo hermanni. The study is based on large sample (more than 3500 failure eggs), from the artificial incubation. Data on embryonic mortality in turtles and tortoises are scarce, thus the data presented in the manuscript are worth to be published. However, I think the version of the manuscript should be improved. I hope that the following comments can be used to improve the manuscript.
General comments:
I feel that the manuscript could be arranged in a better way. taken into account and changed
For example, the Introduction section:
- the Authors start (the 1st paragraph of the manuscript) from “The main threats to the survival of Hermann’s tortoise…”, but basic information about the species are presented in the second paragraph (“TH is a medium-sized terrestrial species that occurs naturally in the European Mediterranean. (…)”.
Paragraph 1 was moved to “Discussion”.
- Information that about the yolk content in reptiles, and that reptile eggs do not contain chalaza, are important for the study. However, in the version of the text, such information are in the 3rd paragraph of the Introduction, just between information about size and distribution of the species (paragraph 2.), and information about the tortoise ecology/activity (paragraph 4.).
Aubert et all (2015) reference has been added but as a fresh egg was not the object of our investigation we believe that more explanation is not necessary. We have stressed some sentences in the text as well.
It was difficult, for me, to follow the Discussion section. The text of the section is divided by photos. Additionally, in the captions of the photos, there are a lot information about results(?) For example: lines 169-170, from caption of the Figure 3 “The weight of the tortoise and its average egg weight were compared. No statistically significant correlation was between tortoise weight and average egg weight laid (r = 0.124, p = 0.459).” Forgive me for my possible misunderstandings, but I think that results from other studies (i.e., from literature), and results from this study, are mixed in the figure captions; it could be confusing for readers. [By the way: the mentioned results of the statistical analyses is incomplete: there is no information on the sample size. There is no information about the statistical analyses in the Materials and methods section, either.]
Photos are really good. However, in such version, they are rather good for ‘supplementary materials’, I think. In the text, the photos/figures are cited only once: line 125 “The egg failures and the causes of embryonic mortality are shown in Figures 1 to 19.” Thus, or the figures are not essential for the text (then can be publish as ‘supplementary materials’), or the text and figures in the Results and Discussion sections should be arranged in an another way.
To recapitulate: I recommend to try to arrange the text in a better way.
The text has been arranged.
Additionally, in the Discussion section I would like to see more important for the study information, for example: if such proportion of egg failure is typical for tortoises? // or typical for the studied species? // for other reptiles? Or, maybe, there is lack of such data for tortoises? Another important question: if the results are typical for eggs from farms, but for ‘wild’ populations situation is different?
There is lack of such data, this has been mentioned additionally in “Discussion”
To say it short: I am not expert in the area (I believe, Authors are such experts), and without good discussion I have problems with evaluating significance of the collected data. taken into account and changed
Several specific comments:
I am not sure, if using the abbreviations “TH”, “THH”, “THB”, is good for the text… However (if Authors will decide to leave the abbreviations), I recommend not to use such abbreviations in the figure and table captions. Typically, in scientific papers, tables and figures should have self explanatory captions, i.e., captions should provide sufficient information to the readers to understand the table/figure without looking for information in the text. Thus, name of the studied species is crucial in any figure/table caption.
Taken into account and changed to THB = T. h. boettgeri, THH = T. h. hermanni in TH = T. hermanni. In the figures full name is used.
The Table 1 is difficult to understand, and should be arranged in a better way.
Table has been deleted, all data are in the text.
lines 76-79. I am not sure why just the literature position have been cited here: there are cited papers on soft-shelled turtles, marine turtles, Emydid turtles… However, for example, one of the the most classical paper in the subject is not mentioned (Yntema C. L., 1968. A Series of Stages the Embryonic Development of Chelydra serpentina. J. Morph. 125: 219-252).
This authors have been added but they are describing a water turtle so no correlation could be found.
I understand that is no possible (and no necessary!) to cite all papers, but in the version it looks like “a short literature review” in the subject, i.e., on embryonic development. But what about tortoises? The study concerns on Testudo species. If there are no data on embryonic development of tortoises – write down it.
There are no data on embryonic development of tortoises.
Abstract:
-- results presented in the abstract are not easy to read,
-- there is lack of summary of the study, in the Abstract.
Results has been added.
I found no sufficient information on methodology of the study, for example, on procedures during handling and transporting eggs. It is important for the study, as e.g., “In the case of incorrect movement or rotation of the egg, the embryonic vessels may rupture and consequently cause embryonic death [7].” lines 61-63.
Explanation written in “Material and Methods”.
lines 106-107: “The temperature at hatching varied between 31 and 32 °C.” Why such temperature were used? As “Temperature-dependent sex determination is characteristic of TH” (lines 72-73), I am not sure if the used incubation conditions are the best ones.
The farm owner wants to obtain more females so he is using hatching temperature between 31 and 32 °C.
lines 112-115: “The embryos were classified into three groups of embryonic development. (…) a maximum of 1.0 cm in length, (…) in size from 1.0 cm to a maximum of 1.5 cm, and (…) longer than 1.5 cm.”
thus: “embryonic development” or just “embryos of different sizes”(?)
We have changed group classification according to different embryos sizes.
Thank you for your suggestions.
Sincerely yours author and coauthors.
Alenka DovÄŤ

Round 2
Reviewer 1 Report
This is a vastly improved manuscript. Thank you for your diligence. Very easy to read now.
For lines 201, 247 and 272 the word data should be used as plural. For example "literature data are" or "There are no data". Alternative "the datum is" .
Author Response
Dear reviewer!
We changed words in plural form.
Thank you for all your suggestions.
authors
Reviewer 3 Report
The manuscript has been significantly improved.
I still feel that the manuscript could be arranged in a better way, but this version is much better than previous one.
There are a lot of photos in the manuscript. It should be decision of the Editor, if all of them will be in the manuscript, or they (part of them) will be moved to supplementary materials.
I have one more remarks for the study (I have written a little about this in the previous review, but I feel it could be important for the future):
“Temperature-dependent sex determination is characteristic of Testudo hermanni. Sex ratio of 50% for T. h. boettgeri is at threshold temperature 31.5°C[7]”. (lines 76-77) and
“The temperature at hatching varied between 31 and 32 °C.” (lines 109-110).
However, there are published some good works which suggest that TSD is adaptive, and incubation temperatures influence fitness, e.g. fitness of males is greatest for males that hatch from eggs incubated at temperatures that naturally produce males (not for males that hatch from eggs incubated at the pivotal temperature, i.e. the temperature at which the sex ratio is about 50%). Thus, I am not sure if “The temperature at hatching varied between 31 and 32 °C.” is the best one for such hatchery. But it is just to think over, for the future works.
Author Response
Dear reviewer!
Thank you for all your suggestions also about a threshold temperature.
authors